# Hemagglutinin Antibodies in the Polish Population during the 2019/2020 Epidemic Season

**DOI:** 10.3390/v15030760

**Published:** 2023-03-16

**Authors:** Karol Szymański, Katarzyna Kondratiuk, Ewelina Hallmann, Anna Poznańska, Lidia B. Brydak

**Affiliations:** 1National Influenza Center, Department of Influenza Research, National Institute of Public Health NIH—National Research Institute, 00-791 Warsaw, Poland; 2Department of Population Health Monitoring and Analysis, National Institute of Public Health NIH—National Research Institute, 00-791 Warsaw, Poland

**Keywords:** influenza, vaccine, antibody, virus

## Abstract

The aim of the study was to determine the level of antibodies against hemagglutinin of influenza viruses in the serum of subjects belonging to seven different age groups in the 2019/2020 epidemic season. The level of anti-hemagglutinin antibodies was tested using the hemagglutination inhibition (HAI) test. The tests included 700 sera from all over Poland. Their results confirmed the presence of antibodies against the following influenza virus antigens: A/Brisbane/02/2018 (H1N1)pdm09 (48% of samples), A/Kansas/14/2017/ (H3N2) (74% of samples), B/Colorado/06/ 2017 Victoria line (26% of samples), and B/Phuket/3073/2013 Yamagata line (63% of samples). The level of antibodies against hemagglutinin varied between the age groups. The highest average (geometric mean) antibody titer (68.0) and the highest response rate (62%) were found for the strain A/Kansas/14/2017/ (H3N2). During the epidemic season in Poland, only 4.4% of the population was vaccinated.

## 1. Introduction

Seasonal flu is a respiratory illness caused by a highly contagious virus [1]. It can spread from person to person in several ways: droplets together with respiratory secretions (coughing, blowing the nose, talking), direct contact, and airborne or indirect contact (contaminated surface) [2,3]. There are three types of human pathogenic influenza virus: A, B, and C, but only A and B are considered clinically relevant. Influenza A viruses consist of numerous subtypes that are divided based on the variety and combinations of glycoproteins found on the viral surface: hemagglutinin (HA), which allows the virion to anchor to the cell surface, and neuraminidase (NA), which allows the virus to be released from host cells [4]. As of now, there are 18 types of hemagglutinin and 11 types of neuraminidase. Types linked with humans are: H1, H2, H3, H5, H6, H7, H9, and H10 [5,6]. Influenza B viruses also contain those glycoproteins on their surface, but they are not divided into subtypes. It was not until the late 1980s that two lines of influenza B virus were isolated: Victoria and Yamagata [7,8]. Hemagglutinin and neuraminidase are antigens, which means they are recognized by the immune system and can trigger an immune response [9].

Due to the segmental nature of the genetic material, the influenza virus is highly mutagenic. Two types of genetic changes can be distinguished: antigenic shift and antigenic drift. Antigenic drift occurs as a result of a point mutation in genes, leading to an altered sequence of amino acids that changes the antigenic site. This is caused by seasonal influenza epidemics [10]. An epidemic is said to occur when the number of cases of a given disease within a specific area and at a specific time is clearly higher than in previous years. In the northern hemisphere, in a temperate climate, influenza epidemics usually occur in winter months [11]. In contrast, antigenic shift takes place when several viruses infect the same host cell and then exchange segments of genes encoding hemagglutinin and neuraminidase with each other. This gives rise to a new virus with new gene constellations, which may cause a pandemic [12]. Pandemics occur when a previously uncirculated influenza virus emerges, causing numerous illnesses due to the lack of immunity in the population and spreading over large areas [1].

Due to its structure, influenza A virus is most susceptible to antigenic variability and is the most common cause of pandemics [13], while influenza B virus causes seasonal epidemics. Influenza C infection is common, but is usually asymptomatic or causes a mild respiratory disease [1].

Due to seasonal influenza epidemics and the risk of a new pandemic, the virological and epidemiological monitoring of influenza virus is of fundamental importance. Currently, there are 149 National Influenza Centers (NICs) in 123 countries as part of the Global Influenza Surveillance and Response System (GISRS) [14].

Any change to the flu virus lowers the chance of a successful immune response. Therefore, it is important to immunize the body every season with a vaccine that contains virus strains selected based on virological data collected in a given hemisphere by the National Influenza Centers [15]. The composition of the flu vaccine must be updated regularly due to the variability of the virus, so it is important to take it every season. There are several methods of influenza virus diagnosis, e.g., virus isolation in chicken embryos; cell culture; antigen detection by IF, ELISA, RT-PCR, rRT-PCR; bedside tests; and serological methods; e.g., HAI [16]. The hemagglutination inhibition (HAI) assay is widely used to evaluate the antibody responses induced by the vaccine as well as for the antigenic characterization of influenza viruses. This is a conventional method used in various aspects of global influenza surveillance, diagnosis, antigen characterization, and vaccine evaluation [17].

Hemagglutination is a process whereby the surface glycoprotein—hemagglutinin—binds to the sialic acid sites on the surface of red blood cells (RBCs), thus creating a stable suspension of RBCs in solution. Anti-HA antibodies bind to hemagglutinin, blocking the possibility of hemagglutinin binding to RBCs, and as a result, hemagglutination is inhibited. By testing successive dilutions of a patient’s serum, antibody levels can be measured [18]. Due to its simplicity, the HAI test has long been used to detect virus-neutralizing antibodies in serum. It is assumed that a titer ≥ 40 reduces the risk of influenza by 50% and is referred to as a seroprotective titer [19]. The titer of hemagglutination-inhibiting antibodies is currently the main immunological marker correlated with protection against influenza [20].

We are focusing on antihemagglutinin antibodies, because they are produced during the viral infection or three to six weeks after the vaccination and are responsible for blocking virus ability to adsorb to host cells. The viral hemagglutinin protein plays crucial role in the process of infecting its host. Hemagglutinin is responsible for membrane fusion, entry of virion to cell, and binding to host receptors. Approximately 80% of all viral envelope proteins are hemagglutinins.

This study was conducted to determine the average levels of antibodies and protective titers against influenza viruses in the Polish population. The results of the study provide an insight into the course of the season in terms of serology, which is also important for influenza surveillance in Poland because not every patient potentially exposed to influenza viruses had a PCR test. Without a swab sample from such patients, we are unable to determine whether they actually had contact with the influenza virus or just flu-like viruses with similar symptoms.

Determining the level of antibodies and determining whether the patient had a protective titer gives information about contact with the virus. Values below the protective titer indicate the lack of contact with the influenza virus, or if the person has been vaccinated, whether he is able to respond to the vaccine or not. When there are no antibodies after vaccination, the person is “non-respondent”. The level of antibodies after the vaccination is decreasing during the epidemic season. It is possible to vaccinate a second time to achieve full protection for the duration of the season. 

## 2. Materials and Methods

Sera were obtained from patients belonging to 7 age groups (0–4, 5–9, 10–14, 15–25, 26–44, 45–64, and 65 years or older) as those age groups are used in Poland in influenza surveillance and in reporting influenza cases to epidemiological departments. Samples were collected at voivodeship sanitary and epidemiological stations (VSES) in Poland, between 1 October 2019, and 31 March 2020. Anonymized samples were then sent to the Influenza Research Department, National Influenza Center. Until the test was performed, the samples were stored at −30 °C.

In each age group, 100 serum samples were tested—700 samples in total. Samples were selected from each VSES to maintain representability for each region and checked for hemolysis in the sample. If noticed, sample was discarded. The hemagglutination inhibition (HAI) test was used to determine the antibody level. All viruses are corresponding to those used in flu vaccine for the 2019/2020 season.

Influenza viruses used in research:Subtype A/H1N1/: A/Brisbane/02/2018Subtype A/H3N2/: A/Kansas/14/2017Influenza type B, Victoria lineage: B/Colorado/06/2017Influenza type B, Yamagata lineage: B/Phuket/3073/2013

All viruses were obtained from World Influenza Centre at Francis Crick Institute, London, and then propagated in chicken embryos in NIC in Poland. Then titer of each virus was determined. Labelled vials containing viruses were stored at −80 °C upon using in research. 

Necessary viruses with high titer were selected to be used in this test. After thawing the viruses, their titer was checked once again on the day of performing the test. Then, a solution of titer 1:8 from each of the virus was prepared. For example, for virus titer 1:64, suspension of the virus was diluted 8 times. For 1:16—two times. After each dilution, the titer was checked, as it is important to use 1:8 titer only. After preparing all necessary solutions, they were stored at 4 °C upon adding them on the plates. 

PBS and 0.1% calcium salt are prepared in–house. In this study, V-bottom, clear, microtitration plates were used. 

Preparation of Receptor Destroying Enzyme (RDE) (Sigma-Aldrich, Jerusalem, Israel IL). Lyophilisate of RDE is resuspended in 5 mL of sterile water, and mixed firmly, then filled up to 100 mL with 0.1% of calcium salt of pH = 7.4. Then, 150 µL of prepared solution is pipetted into a sterile tube and stored at −30 °C until required. 

The study used chicken red blood cells. Blood cells delivered to the laboratory were suspended in Alsever’s solution. Alsever‘s solution is prepared in-house. In order to obtain packed cells, 5 mL of cells were collected from Alsever’s solution into a 50-mL centrifuge tube, topped up to 50 mL with PBS, then centrifuged for 10 min at 1200 RPM. After centrifugation, the supernatant was decanted, topped up again to 50 mL with PBS, and then washing and centrifugation at 1200 RPM for 5 min were repeated three times. In the next step, the centrifuged blood cells were transferred to a new 15-mL centrifuge tube and filled with PBS to 12 mL, then centrifuged for 10 min at 1200 RPM. The packed blood cells obtained in this way were used in further studies. In order to obtain the appropriate blood cell concentration for the HAI test, the following proportion was used: 1 mL PBS:5 µL packed cells. 

Each of the sera was treated with RDE for 16 h at 37 °C prior to the hemagglutination inhibition test. For this purpose, 50 µL of serum from the patient was added to 150 µL of RDE, followed by incubation under appropriate conditions. After this step, to inactivate the enzyme, the mixture was incubated at 56 °C for 30 min.

In order to eliminate non-specific hemagglutination inhibitors, a mixture of blood cells and PBS was prepared in a ratio of 1 volume of packed blood cells to 20 volumes of serum after incubation with RDE. The serum-cell mixture was incubated for one hour at 4 °C. After incubation, the suspension was centrifuged at 1200 RPM for 10 min—the supernatant contained serum ready for further processing.

To row A, 50 µL of the patient’s prepared serum was pipetted, and then a serial dilution was made in PBS. Then, the prepared solution of the virus, which has a titer of 1:8, was added to each well on the plate. After virus addition, the plate was incubated for 15 min at room temperature. After incubation, 50 µL of blood cell solution was added. Readings were taken after 30 min of incubation at room temperature. 

The plate was lifted and tilted to allow RBCs to run down to the side of the well. Readings were performed according to pattern showed in the Figure in the Appendix A. 

The analysis of the results for each of the virus subtypes was carried out in 7 age groups. Numbers and percentages of patients with adequate antibodies and patients achieving a protective level (antibody titer ≥ 40) were determined. The average antibody level in the study group was calculated as a geometric mean of non-zero values (GMT) with 95% confidence intervals [CI]. The following statistical tests were employed in the analysis: chi-square test (for comparisons between age groups and epidemiological seasons in terms of antibody occurrence frequency and reaching protective level); Mann–Whitney test (comparison of antibody titer distributions between two epidemiological seasons); Kruskal–Wallis test (comparisons of antibody titer distributions between 7 age groups and 3 epidemiological seasons—post-hoc tests with Bonferroni correction for multiple comparisons were used to identify differing seasons). A significance level of 0.05 was assumed in all of the analyses.

Calculations were performed using the SPSS 12-PL statistical software.

## 3. Results

The serum of people belonging to all the age groups showed the presence of antibodies against all the four viruses analyzed. In total, 48% of the subjects had antibodies against the H1 subtype, H3—74%, B/Colorado—26%, and B/Phuket—63%. Figure 1 and Figure 2 show the percentage of patients in individual age groups that had antibodies against a particular influenza virus.

Using the chi-square test for analysis, statistically significant differences in the prevalence of antibodies (between seven age groups) were found for the following influenza virus types and subtypes:For subtype H1: *p* = 0.001, the proportion of subjects with antibodies ranged from 33% in the group of 0–4 years to 60% in the group of 10–14 years.For the Victoria line (B/Colorado): *p* < 0.001, the proportion of subjects with antibodies from 7% in the group of 15–25 years to 47% in the group of 65 years or more.For the Yamagata line (B/Phuket): *p* < 0.001, from 27% of subjects with antibodies in the 45–64 group to 85% in the 0–4 age group.

The analysis showed no statistically significant differences for the H3 subtype, the percentage of subjects with antibodies ranged from 68% in the 10–14 age group to 82% in the 5–9 group.

Statistically significant differences in the percentage of subjects with antibodies between ages up to 14 and over 14 were observed only for antibodies against B/Colorado and B/Phuket (analysis using the chi-square test):For B/Colorado *p* = 0.015, 22% of subjects over 14 years of age and 30% of children up to 14 years of age had antibodies.For B/Phuket *p* = 0.007, 59% of subjects over 14 years of age and 69% of children up to 14 years of age had antibodies.For the H1 subtype: 46% of subjects over 14 years of age and 50% of children up to 14 years of age had antibodies.For the H3 subtype: 74% of subjects over 14 years of age and 75% of children up to 14 years of age had antibodies.

Figure 3 shows geometric mean antibody titers (GMT) in the sera of patients who had antibodies. A statistical analysis using the Kruskal–Wallis test showed a statistically significant difference between the seven age groups for all the antibody types:For the H1 subtype: 335 people had antibodies, average level 54.4, statistical significance of differences—*p* = 0.002, the lowest level of antibodies in the group of 26–44 years—GMT = 40.0 [CI = 33.9–46.1] the highest in the group of 10–14 years—GMT = 88.8 [CI = 81.1–96.4].For the H3 subtype: 521 people had antibodies, average level 68.0, [CI = 65.9–70.1]. *p* = 0.004, the lowest level in the age groups 26–44 and 45–64 years of age—GMT = 55.5, [CI = CI: 49.9–61.2 and 49.8–61.2, respectively], the highest in the 5–9 age group—GMT = 89.3 [CI: 83.9–94.7].For the Victoria (B/Colorado) line: 180 people had antibodies, average level 25.3, [CI: 21.8–28.7], *p* < 0.001, the lowest level in the group of 26–44 years—GMT = 14.5 [CI: 4.5–24.5] and the highest in children up to 4 years of age—GMT = 41.3 [CI = 33.4–49.3].For the Yamagata line (B/Phuket): 443 people had antibodies, average level 40.9 [CI = 38.8–43.1], *p* < 0.001, the lowest level in the group of 46–64 years old—GMT = 23.3 [CI = 15.1–31.5], the highest in children up to 4 years of age—GMT = 51.9 [CI = 47.5–56.4].

Figure 4 shows the response rates (percentages with protective anti-HA titers ≥ 40) for the four tested influenza viruses in seven different age groups. In each of the groups, subjects with a protective titer of anti-HA antibodies ≥ 40 were found. Statistical analysis (using the chi-square test) showed a statistically significant difference in the protection factors between the seven age groups for three subtypes. In the case of H3, the difference was on the border of statistical significance.
For subtype H1: 239 people had antibodies with a titer ≥ 40, the protective factor was 34%; statistical significance of differences between age groups—*p* < 0.001 (rates from 20% in the 0–4 age group to 51% in the 10–14 age group).For subtype H3: 431 people had antibodies with a titer ≥ 40, protective factor was 62%; statistical significance of differences between age groups—*p* = 0.082 (rates from 55% in the 45–64 and ≥65 age groups to 72% in the 5–9 age group).For the Victoria line (B/Colorado): 79 people had antibodies with a titer ≥ 40, protective factor was 11%; statistical significance of differences between age groups—*p* = 0.001 (rates from 2% in the 15–25 group to 24% in the 0–4 age group).For the Yamagata line (B/Phuket): 297 people had antibodies with titers ≥ 40, protection factor was 42%; statistical significance of differences between age groups—*p* = 0.001 (rates from 7% in the group of 45–64 and ≥65 years to 68% in the group of 0–4 years).

For influenza B virus of the Yamagata lineage, which remained unchanged in the vaccine composition in the epidemic season from 2017/2018 to 2019/2020, an analysis of differences in antibody levels between the three seasons was performed. The frequency of reaching the protective level (using the chi-square test) and the distribution of antibody titers were compared.

There were statistically significant differences between the three compared seasons in the percentage of subjects with antibodies at the level ≥ 40 (*p* < 0.001), with the highest value recorded in the 2018/19 season (67%) and the lowest in 2017/18 (34%) (Table 1).

Statistically significant differences occurred in six out of seven analyzed age groups (*p* < 0.001 in all cases), with the exception of the 26–44 group where the percentages amounting to 40%, 47%, and 51% did not show statistically significant differences.

The lowest percentages mostly concerned the 2017/18 season; lower values were recorded in the 2019/2020 season only in the two oldest age groups. The highest percentages usually concerned the 2018/2019 season; higher values were observed in the 2019/2020 season only for two age groups. The exceptions are as follows: the 0–4 age group and the 26–44 age group; in the latter case, however, the differences between the seasons were not statistically significant (Table 1).

Mean antibody values (GMT) by season and age group as well as the statistical significance of differences between the three seasons are shown in Table 1. Statistically significant differences in terms of the level of antibodies against B/Phuket were observed in four age categories (0–4 10–14, 15–25, and 26–44 years old) and for all groups together. The analysis used the Kruskal–Wallis test supplemented with post-hoc tests to identify differing pairs: In the 0–4 age group, the seasons 2017/2018 and 2019/2020 differ statistically significantly, in which the average level of antibodies was 54.4 and 69.4, respectively (*p* = 0.008 after adjusting for multiple comparisons using the Bonferroni method).
In the 10–14 age group, the level of antibodies in the 2018/19 season (average 133.7) significantly differed from the values in the other seasons (54.6 and 51.2), in both cases *p* < 0.001 (even after adjusting for multiple comparison by means of Bonferroni method).In the 15–25 age group, the level of antibodies in the 2017/2018 season (47.6) was significantly different from that observed in 2018/19 (62.2, *p* = 0.020, after taking into account the correction) and 2019/2020 (62.7, *p* = 0.047, after correction),In the 26–44 age group, the level of antibodies in the 2019/2020 season (89.2) was significantly different from that observed in 2017/18 (57.2) and 2018/19 (58.3)—in both cases *p* < 0.001, after the correction.

Taking an overall look at all the age groups, the level of antibodies in the 2017/2018 season (58.3) differed both from the one observed in the 2018/19 season (70.9, *p* < 0.001 after adjustment) and in the 2019/2020 season (65.0, *p* = 0.011, after correction).

An analysis of differences in the level of antibodies against influenza B virus of the Victoria line, which remained unchanged in the composition of the vaccine in the epidemic seasons from 2018/2019 to 2019/2020, was also carried out.

Statistically significant differences were recorded between the compared seasons in terms of the percentage of subjects with anti-B/Colorado antibodies ≥ 40 (chi-square test, *p* < 0.001)—the protection factor was 22% in the 2018/19 season compared to 11% in 2019/20 (Table 2). This effect consisted of significantly higher percentages in the 10–14, 15–25, and 26–44 age groups in the 2018/2019 season. In the oldest and youngest age groups, the differences were statistically insignificant (Table 2).

There were no statistically significant differences between the season 2018/2019 and 2019/2020 in the average level of antibodies against B/Colorado (Mann–Whitney test)—the average titer was 52.9 vs. 55.8, respectively (Table 2). Such differences occurred only in two age categories in children: the 0–4 age group, in which a significantly higher level of antibodies was recorded in the 2019/2020 season (77.7 vs. 48.1; *p* = 0.008), and the 10–14 age group with a significantly higher level in the 2018/2019 season (50.6 vs 40.0; *p* = 0.004).

## 4. Discussion

Antibodies against all four tested viruses were found in patients of all age groups. All the viruses analyzed were included in the influenza vaccine for the 2019/2020 season. However, due to the low vaccination rate in the population (only 4.4% of Poles received the flu vaccine in the 2019/2020 season) [21]. In comparison, the vaccination rate calculated from administered doses in the past seasons were characterized by an even lower rate: 2018/2019: 3.9%, 2017/2018: 3.6%. It should be assumed that the presence of antibodies in the serum is the result of contracting the disease.

A protective antibody titer is considered to be ≥40. High titers of antibodies may indicate a past infection. Immunoglobulin levels also increase after influenza vaccination. In the 2019/2020 epidemic season, the highest percentages of patients with a protective antibody titer were observed for virus A/Kansas/14/2017 (subtype A/H3N2/), and the lowest for B/Colorado/06/2017, although this strain has been in the vaccine since the season 2018/2019.

In the case of the influenza A virus, the A/H3N2/ subtype featured the highest percentage of subjects with a protective level of antibodies, despite the fact that this season the most molecularly confirmed strain was the A/H1N1/pdm09 subtype.

The main role of the influenza vaccine is to increase immunity against influenza virus infection. Vaccination is an effective method of preventing the disease and its complications and related mortality. The effectiveness of the influenza vaccine varies depending on the patient age, the effectiveness of the immune system response, as well as on the match between the circulating strains and the vaccine strains in a given year [6]. However, cross-immunity, i.e., alleviation of disease symptoms, can be observed even though the vaccine received was for a different variant of the virus [22], which does not occur in young children [23]. On the other hand, vaccination is more effective in children than in the elderly. Therefore, vaccinating children may be an important way to prevent the disease and post-disease complications in elderly people [24], whose immune response is weakened [20].

As research conducted in Poland in the period from December 2018 to April 2020 among the examined patients shows, influenza was found mainly in unvaccinated subjects [25]. Therefore, taking the flu vaccine every season is recommended not only for people belonging to the high-risk group. Seasonal influenza vaccination is also necessary because immunity declines over time after vaccination [26]. In addition, a new disease caused by the SARS-CoV-2 virus appeared in the 2019/2020 epidemic season. This virus has a similar route of transmission and causes a disease with similar symptoms [27,28]. That is why a correct diagnosis is important to ensure that patients are adequately treated. However, we did not notice any impact of wearing masks or other restrictions on antibody levels. It can be explained by the fact that restrictions in Poland were incorporated in April of 2020. Testing patients’ sera is highly connected with the vaccination policy, as it informs us about immunity against influenza viruses in the population. Based on those findings, we can recommend actions to ministry of health of Poland. 

Low protection rates confirm the low vaccination rate in Poland. The vaccination rate should be higher for the elderly and people at high risk; the vaccination rate recommended by the WHO should reach 75%. To achieve that, influenza vaccines should be mandatory for children up to 14 years of age, the elderly above 65 years of age, and all individuals in risk groups. 

## Figures and Tables

**Figure 1 viruses-15-00760-f001:**
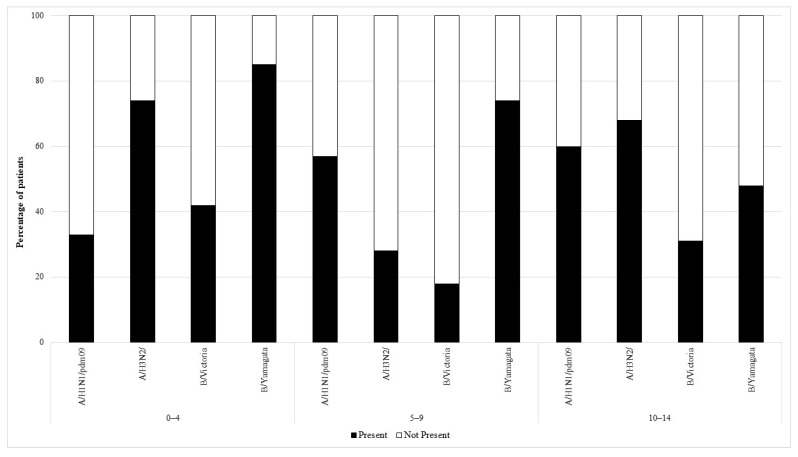
The presence of antibodies in the serum of patients aged 0–4 years, 5–9 years of age, and 10–14 years of age in the 2019/2020 epidemic season.

**Figure 2 viruses-15-00760-f002:**
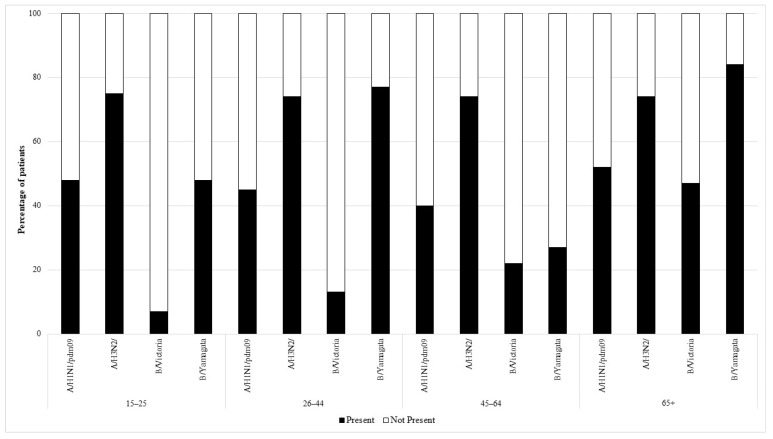
The presence of antibodies in the serum of patients aged 15–25, 26–44, 45–64, and 65+ years of age in the 2019/2020 epidemic season.

**Figure 3 viruses-15-00760-f003:**
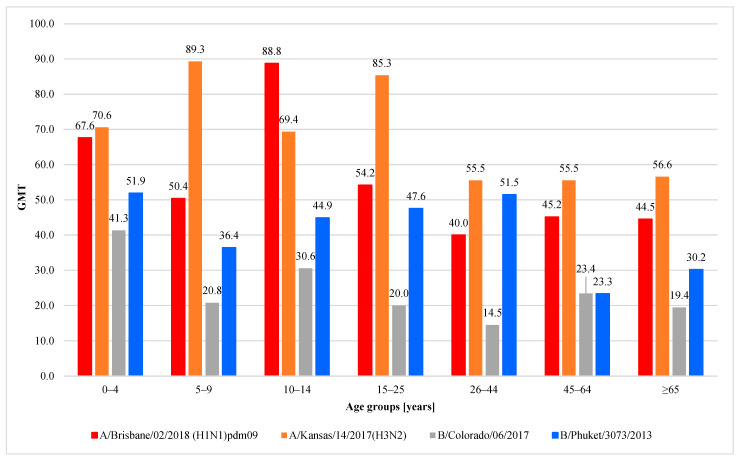
Geometric mean titers of anti-haemagglutinin antibodies (GMT) in the epidemic season 2019/2020 in age groups in Poland.

**Figure 4 viruses-15-00760-f004:**
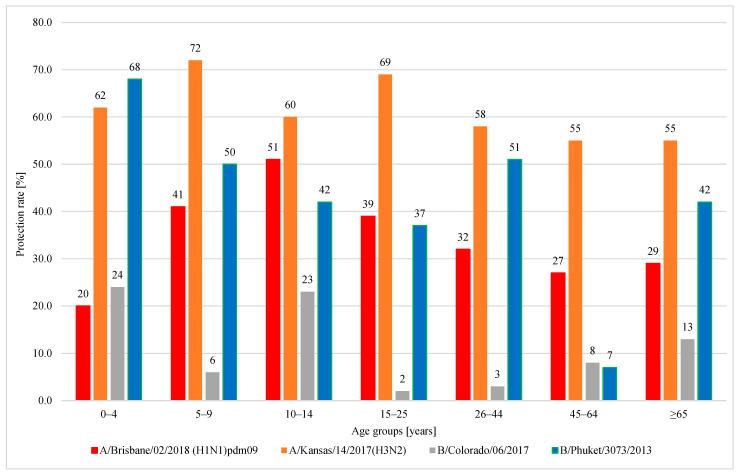
Percentage of cases with a protective titer of anti-haemagglutinin antibodies (%) in the 2019/2020 epidemic season in different age groups.

**Table 1 viruses-15-00760-t001:** Comparison of three epidemic seasons for B/Phuket/3073/2013—Yamagata Lineage.

**Age Group** **[Years]**	**Protection Rate**			**The Significance of the Differences (*p*-Value)**
	**2017/2018**	**2018/2019**	**2019/2020**	
0–4	41%	55%	68%	<0.001
5–9	31%	60%	50%	<0.001
10–14	27%	93%	42%	<0.001
15–25	27%	68%	37%	<0.001
26–44	40%	47%	51%	Not Significant (NS)
45–64	25%	71%	7%	<0.001
>65	48%	75%	42%	<0.001
Total	34%	67%	42%	<0.001
**Age Group** **[Years]**	**Antibodies Level (GMT)** **with 95% Confidence Intervals**			**The Significance of the Differences (*p*-Value)**
	**2017/2018**	**2018/2019**	**2019/2020**	
0–4	54.4 [50.6–58.1]	64.9 [61.0–68.8]	69.4 [65.5–73.2]	0.011
5–9	63.2 [58.1–68.3]	52.8 [49.8–55.8]	49.9 [46.1–53.8]	NS
10–14	54.6 [49.2–60.1]	133.7 [130.2–137.2]	51.2 [46.9–55.6]	<0.001
15–25	47.6 [43.2–51.9]	62.2 [58.9–65.6]	62.7 [57.3–68.1]	0.015
26–44	57.2 [52.9–61.5]	58.3 [54.4–62.2]	89.2 [84.1–94.3]	<0.001
45–64	49.2 [44.2–54.1]	57.5 [54.5–60.5]	65.6 [49.6–81.6]	NS
>65	76.1 [71.6–80.7]	67.6 [64.1–71.1]	71.3 [65.2–77.4]	NS
Total	58.3 [56.5–60.1]	70.9 [69.4–72.3]	65.0 [63.0–67.0]	<0.001

Note: NS: Not Significant.

**Table 2 viruses-15-00760-t002:** Comparison of two past epidemic seasons for B/Colorado/06/2017—Victoria Lineage.

**Age Group** **[Years]**	**Protection Rate**		**The Significance of the Differences (*p*-Value)**
	**2018/2019**	**2019/2020**	
0–4	23%	24%	NS
5–9	11%	6%	NS
10–14	47%	23%	<0.001
15–25	10%	2%	0.019
26–44	37%	3%	<0.001
45–64	10%	8%	NS
>65	13%	13%	NS
Total	22%	11%	<0.001
**Age Group** **[Years]**	**Antibodies Level (GMT)** **with 95% Confidence Intervals**		**The Significance of the Differences (*p*-Value)**
	**2018/2019**	**2019/2020**	
0–4	48.1 [43.3–52.8]	77.7 [68.8–86.7]	0.008
5–9	41.7 [36.0–47.3]	40.0 [32.0–48.0]	NS
10–14	50.6 [47.0–54.1]	40.0 [35.9–44.1]	0.004
15–25	43.9 [37.4–50.3]	80.0 [43.1–116.9]	NS
26–44	64.6 [59.7–69.4]	40.0 [28.7–51.3]	NS
45–64	52.8 [45.6–60.0]	51.9 [42.0–61.8]	NS
>65	59.8 [51.4–68.2]	68.2 [56.0–80.4]	NS
Total	52.9 [50.7–55.0]	55.8 [51.7–60.0]	NS

Note: NS: Not Significant.

## Data Availability

Not applicable.

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
