# Peer review of "Hemagglutinin Antibodies in the Polish Population during the 2019/2020 Epidemic Season"

_viruses, 2023, doi:10.3390/v15030760_

Round 1

Reviewer 1 Report

The manuscript that I reviewed “Hemagglutinin antibodies in Polish population during the 2019/2020 epidemic season” is a study aimed to assess, by hemagglutination inhibition test, levels of antibodies and protective titers against 4 influenza viruses (A/Brisbane/02/2018 (H1N1)pdm09, A/Kansas/14/2017/ (H3N2), B/Colorado/06/ 2017 Victoria line and B/Phuket/3073/2013 Yamagata line) in the serum of Polish patients, in the 2019/2020 epidemic season. The seroprevalence age distribution for 7 different age groups was performed.

 Major comments

The article is well written and interesting. The statements described are supported by detailed presented data. In my opinion, a weakness of this work is represented by a lack of in-depth of the discussion section that the Authors should improve.

For example, how would the Authors argue the observations reported in line 322-325 and 326-328?

Line 329-339: the reported concepts in my opinion seem not very pertinent or at least additional but not functional to the discussions of the obtained results also considering the study conducted.

Furthermore, in my opinion, it will be useful to add numeric data about the composition of the age groups and detailed information about the collection date as the Authors analyzed also other epidemic seasons.

As the Authors described that a new disease caused by the SARS-CoV-2 virus appeared in the 2019/2020 epidemic season, do they think that data obtained could be influenced by the use of protective masks?

Minor comments:

In the “Introduction” section I suggest to delete one of the two sentences in line 33-34 or 86-87 since it is the same information.

Line 82-84 should be re-worded for better grammatical sense.

Reviewer 2 Report

I read the manuscript entitled "Hemagglutinin antibodies in Polish Population during the epidemic season" and found some major issues. As the study used samples from humans, it is mandatory to use ethical approval but this part is missing. There is no clear sample size calculation procedure for the study. The design of the study is not well explained. The other issues that I identified: 

Introduction: The chronology of the writing is missing and the literature cited is not fitted well.

Methods:

-Add the sample size calculation that used for your study with the appropriate references

-What are the sensitivity and specificity of the test the authors used? What is the justification for the use of the serological test for interpreting the results

-How the authors used the age group? What are the justifications of use this category? Use the references for the age categories with references that you used in the study

-Delete figure 1 as it is a standard procedure for testing 

Discussion:

-Describe only the key findings and avoid the general discussion

Conclusion:

Add a conclusion based on your key/new findings and make a recommendation

Reviewer 3 Report

Reviewer comments on manuscript:

Reference:       Viruses-2218590

Authors:          Karol SzymaÅ„ski, Katarzyna Kondratiuk, Ewelina Hallmann, Anna PoznaÅ„ska, Lidia B. Brydak

Tittle:             Hemagglutinin antibodies in Polish population during the 2 2019/2020 epidemic season

In this article, the research team led by Szymański and Hallmann summarizes the study conducted in a small sample of Polish population aimed at determining the presence of immune response against influenza viruses by measuring the level of anti-hemagglutinin antibodies. The analysis is based on data obtained by applying the HAI test, a simple, sensitive, reliable, inexpensive and rapid method recognized as suitable method for the detection and quantitation of antibodies against influenza virus. Four representative strains were analysed: H1N1 and H3N2 (influenza A) and Colorado2017 and Puket2013 (influenza B). Testing was performed on human serum samples corresponding to 700 patients and age was considered in the analysis (age groups 0-4, 5-9, 10-14, 15-25, 26-44, 45-64 and >65 years). As a result, authors determined that most of analysed samples showed different degrees of acquired immunity to the hemagglutinin antigen of the different variants under study, with statistical variations associated with each age group.

After a reasoned introduction, authors highlight the importance of this kind of studies in the context of epidemiological surveillance of the disease. Subsequent section describes in detail the material and methods used for the determinations and results are further summarised, including graphical support (Figures) and statistical treatment of data evolved. Finally, a brief discussion of the above data completes the work, although without any specific section devoted to establishing final conclusions.

After a careful reading of the manuscript, this reviewer has some major concerns regarding this work. I will subsequently discuss these points as follow bellow:

1-. Introduction.

The introduction is correctly written and exposes the most relevant background of the research, although some aspects could be improved:

1.1-. Considering the local geographic scope of the study (Poland), it would be advisable to add information on the situation of vaccination against influenza in the Polish population in recent years beyond the scarcely commented “…low vaccination rate in the population (only 4.4% of Poles received 317 the flu vaccine in the 2019/2020 season [21]” in the discussion section (page 12, lines 317-318).

1.2-. Paragraph page2, lines 82-88 repeat in some extent data already provided in the same introduction section (i.e. page 1, lines 25-38 H and N subtypes). It could be supressed.

1.3-. Paragraph page 2, line 83. “…absorb?” Please, reconsider this term.

1.4-. Paragraph page 2, line 74. “suspension of RBC”. Reconsider to change (suggested: lattice or aggregate).

1.5-. Paragraph page 2, line 75. “…hemagglutination is NOT inhibited”; really means IS inhibited

2-. Material and methods.

The methodology used in the work is adequate, but it would be eventually necessary to detail or comment:

2.1-. Are the collected samples representative of the territory as a whole or are there some preferential locations? (i.e. largely populated cities vs. less populated rural areas).

2.2-. Gender issues apply? As well as  age distribution has been done, gender could have been another criterion of differentiation of the samples.

2.3-. A justification of the overall sample size and distribution of age groups could be advisable for statistical significance.

2.4-. Please, indicate whether there have been any inclusion/exclusion criteria in the selection of the 100 serum samples analysed belonging to each age group.

2.5-. The HAI method is established as standard and there are many references that describe the methodology in detail. Is such a detailed description necessary in this article (pages 3-4, lines 121-159, including Fig. 1.? If required, these detailed descriptive of the protocol could be moved to a “supplementary material” section apart from the core manuscript.

Results

3.1-. Statistical treatment of data is correct. However, no standard deviation is provided in some cases in order to determine how dispersed the data are regarding the calculated mean (include in tables and Figures).

3.2-. The data shown in page 7, lines 188-197; 204-211; 215-226 should be included in tables, where all values were specified for better understanding.

3.3-. Specify the origin of the "protection rate" data the years 2017/2018 and 2018/2019" that appear in table 1 and 2 of the results section (page 10, page 12).

3.4-. Page 7, lines 201-203. Justify why the data have been analysed in these two age groups separately and the age of 14 years has been chosen for this purpose.

3.5-. Why have no comparisons been made for the "protector rate" in the different years of vaccination for H1N1, H3N2 viruses?

Discussion

Discussion section is very scarce and superficial, merely descriptive of some vague aspects of the study (higher protection detected against A-H3N2, lower against B-Colorado-2017). It does not deepen and compare the results obtained and does not highlight any conclusions apart from the generally assumed regarding the importance of determine the acquired immune protection levels against flu as a consequence of development of the disease. No strengths and weaknesses of the study have been pointed, neither any final concluding remarks.

English Language and style

I am not English native speaker and consequently I don't feel qualified to judge in depth about the English language, but I would suggest to the authors try to carefully check the style of the manuscript and refine some sentences, hard to follow.

Final considerations:

Globally, this contribution contains a remarkable research effort, although this reviewer considers its contents, treatment, novelty and significance do not justify its acceptance. The way the results have been treated and exposed is not clear enough and the associated discussion is very scarce in such a way that, more importantly, no really significant conclusions evolved from it. I feel this manuscript is average in terms of contents, results and originality for the aim and scope of the journal Viruses. Consequently, I recommend rejection of the submission.

In any case, this reviewer recognised that this piece of work could be rewritten and the new manuscript could be interesting for a more specialised reader. Consequently, this contribution could be resubmitted to a more specialised journal focused on epidemiology/public health to be  considered for publication in a more appropriate journal profile.  

Round 2

Reviewer 2 Report

This is very frustrating when the authors avoid the comments from reviewers. I did not see any changes/reflections in the main text of the manuscript. The authors do not care to understand the comments and they did not respond properly as the concern pop up. The ethical approval issue, sample size calculation, sensitivity, and specificity of the test were not responded appropiately. As the manuscript has a major methodological defect, I strongly recommend the rejection of the current form of the manuscript.

Reviewer 3 Report

After a detailed reading of the resubmitted manuscript, this reviewer acknowledges that the proposed article has been substantially improved, making its content clearer and more understandable. Most of the suggested modifications have been incorporated into the work or the authors have reasoned their irrelevance. In this sense, the improved version is formally alligned with what was requested.

However, once the formal objections to the article have been solved, the main concern originally formulated persists in the opinion of this reviewer: the content and results obtained are possibly of much more interest to a reader specialised in epidemiology, so that this work could find a more appropriate niche in a journal focused on that field within the MDPI publishing brand.

Despite the above exposed consideration, this reviewer agrees that the current form of the article meets the minimum requirements for publication. Consequently, if the editor considers that the profile of this piece of research is adequate and fits the scope ande aims of the journal Viruses, it would be accepted.

Very minor comments:

page 4; line 154. Fig 1 refers to Supp Mat figure.
page 6; line 212 (Fig 3=4); page 8, line 233 (Fig 4 = 3) Page 9, line; 255 (Fig5 = 4)  

Author Response

Dear rewiever,

We would like to thank you for your rewievs, we made suggested changes in manuscript.

Best regards,

Authors